# Can SMEs synergistic transformation of digitization and greenization promote enterprise risk-taking?

Juan Wang[1]*, Jianxin Cui[2]

1 School of Accounting, Shandong Women's University, Jinan, Shandong Province, China,
2 Management Engineering Department, Qingdao University of Technology, Qingdao City, Shandong Province, China

* 31013@sdwu.edu.cn

## Abstract

The synergistic transformation of digitization and greenization has become an important direction in China's economic development. Small and Medium Enterprises (SMEs) have the characteristics of small scale and high growth and have become the key force of synergistic transformation of digitization and greenization. Synergistic transformation of digitization and greenization helps to promote the coordination and unification of SMEs' operating efficiency and environmental protection, resulting in the effect of $1 + 1 > 2$, which has become an important factor in promoting enterprise risk-taking. Using the sample of enterprises listed on the SME board from 2012 to 2022, this paper adopts the Haken model to measure SMEs' synergistic transformation of digitization and greenization and analyzes its impact on enterprise risk-taking, drawing the following conclusions: (1) SMEs' synergistic transformation of digitization and greenization significantly promotes enterprise risk-taking. (2) The impact mechanism analysis finds that SMEs' synergistic transformation of digitization and greenization improves enterprise risk-taking by reducing cash holdings and Research and Development (R&D) investment. (3) Further research finds that compared with state-owned SMEs, the synergistic transformation of digitization and greenization of non-state-owned SMEs has a more significant impact on enterprise risk-taking. Compared with high-tech SMEs, the synergistic transformation of digitization and greenization of non-high-tech SMEs has a more significant impact on enterprise risk-taking. These conclusions have practical significance for promoting SMEs' synergistic transformation of digitization and greenization and improving enterprise risk-taking.

## Introduction

The Chinese government is greatly attached to the synergistic transformation of digitization and greenization of enterprises. In March 2021, *the 14th Five-Year*

**Data availability statement:** All relevant data are within the paper and its Supporting Information files.

**Funding:** This paper is funded by "the Shandong Provincial Social Science Planning Research Project", "A Study on the Impact of Corporate Digital Transformation on Risk Taking in Shandong Province (22CGLJ35)".

**Competing interests:** The authors have declared that no competing interests exist.

*Plan for National Economic and Social Development of the People's Republic of China and the Outline of 2035 Vision Goals* clearly stated that the digital economy must be rationally utilized to achieve the goal of *carbon peak* and *carbon neutrality. The 20th National Congress Report* released in October 2022 put forward the concept of synergistic transformation of digitization and greenization for the first time. This determined that the manufacturing industry's intelligent and green coordinated development should be promoted to accelerate carbon reduction, pollution reduction, green expansion, and growth. In November 2022, five departments, including the *Central Cyberspace Administration, the National Development and Reform Commission, the Ministry of Industry and Information Technology, the Ministry of Ecology and Environment, and the National Energy Administration,* announced that they would jointly carry out a comprehensive pilot for the synergistic transformation of digitization and greenization. It explores replicable and scalable experiences in the green and low-carbon development of digital industries and the collaborative transformation of traditional industries. On February 2023, *the Overall Layout Plan for the Construction of Digital China* issued by *the Central Committee of the Communist Party of China* and *The State Council* further clarified that it is necessary to accelerate the synergistic transformation of digitization and greenization. It can be seen that the synergistic transformation of digitization and greenization has become an important development direction of China's economy. Small and Medium-sized Enterprises (SMEs) are the core position in the market economy. Referring to data released by *the National Bureau of Statistics*, SMEs provide more than 60% of GDP, 50% of tax revenue, and 70% of employment opportunities.

With small scale, high growth, and rapid transformation, SMEs actively integrate into the digital economy and green economy development through technological innovation, mode transformation, and industrial upgrading. Therefore, SMEs have become a key force for the synergistic transformation of digitization and greenization. Synergistic transformation of digitization and greenization helps SMEs coordinate resource allocation and realize the coordinated development of digital and green systems. SMEs use digital tools to promote green development, improve production and operation efficiency, promote the coordination and unification of SMEs' efficiency and environmental protection, and produce the effect of *1 + 1 > 2*. In short, SMEs' synergistic transformation of digitization and greenization helps promote the transformation of the economy to low energy consumption, low emissions, and high efficiency. It enhances enterprises' ability to achieve the goal of high-quality economic development.

In the process of digital transformation and green transformation in SMEs, digitization and greenization have mutual influence, and the synergistic development trend is increasingly obvious. In one sense, enterprise digital transformation helps promote green transformation. Moreover, it is proved that enterprise digital transformation promotes green transformation from the perspective of social technology system theory [1]. Green technology innovation is an important means of realizing enterprise green transformation. Enterprise digital transformation alleviates

financing constraints and attracts government subsidies [2–4], promotes R&D expenditure, exerts a spillover effect, forms an innovation cooperation network [5], all of which improve green technology innovation. However, in another sense, enterprise green transformation plays a leading role in the process of digital transformation. The greenization of enterprises requires a wider range of data with larger data types and quantities. An enterprise must expand its digital transformation to acquire more comprehensive data processing platforms, deeper data mining technologies, wider information, improved energy efficiency, precision in green technology innovation, and other benefits to support green transformation.

Enterprise risk-taking is the motivation to pursue additional returns. Only by taking risks can enterprises obtain excess returns. The alteration of the surrounding environment causes the change of risk-taking behaviors. Nonetheless, a good and stable external economic environment is the guarantee for enterprises to dare to bear risks [6,7]. Enterprises actively carry out digital and green transformation under the influence of the digital economy and green economy. Note that existing research shows that both digital and green transformations help promote enterprise risk-taking. Digital transformation improves enterprise risk-taking by easing financing constraints [8] and improving resource allocation efficiency [9]. Otherwise, green credit policy [10,11] and enterprise green mergers and acquisitions [12] improve enterprise risk-taking. However, what is the impact of SMEs' synergistic transformation of digitization and greenization on enterprise risk-taking? How do cash holdings and R&D investment affect the relationship between SMEs' synergistic transformation of digitization and greenization, and enterprise risk-taking? Clarifying the above issues has certain significance for promoting SMEs' synergistic transformation of digitization and greenization, improving enterprise risk-taking, and thus promoting the high-quality development of enterprises.

Therefore, using enterprises listed on SME boards from 2012 to 2022 as samples, this paper analyzes the impact of SMEs' synergistic transformation of digitization and greenization on enterprise risk-taking, which draws the following conclusions: (1) SMEs' synergistic transformation of digitization and greenization has a significant promotion effect on enterprise risk-taking. (2) The mechanism analysis finds that SMEs' synergistic transformation of digitization and greenization improves enterprise risk-taking by reducing cash holdings and R&D investment. (3) Further research finds that compared with state-owned SMEs, non-state-owned SMEs' synergistic transformation of digitization and greenization significantly impacts enterprise risk-taking. Compared with high-tech SMEs, non-high-tech SMEs' synergistic transformation of digitization and greenization significantly impacts enterprise risk-taking.

This study potentially makes the following marginal contribution: (1) The Haken model is the first to be used in this paper to measure the degree of synergistic transformation of digitization and greenization. Existing studies mostly use the Haken model to measure the degree of synergism of multiple macro subsystems [13,14]. To further research, this study uses the Haken model to measure the degree of synergism between an enterprise's digital and green subsystems. This paper expands the application field of the Haken model. (2) This paper empirically tests the impact of SMEs' synergistic transformation of digitization and greenization on enterprise risk-taking. The existing literature finds that digital transformation [8,9,15] and green transformation [16,17] help to promote enterprise risk-taking. Unlike previous studies, this paper takes enterprises listed on the SME board as the research object, analyzing the impact of the synergistic transformation of digitization and greenization of enterprises with strong growth in risk-taking. It enriches the research on the economic consequences of the synergistic transformation of digitization and greenization. (3) This paper explores the mechanism through which the SMEs' synergistic transformation of digitization and greenization influences enterprise risk-taking. This study discovers that SMEs' synergistic transformation of digitization and greenization improves enterprise risk-taking by reducing cash holdings and R&D investment. (4) This paper provides empirical evidence of the economic consequences of the synergistic transformation of digitization and greenization of different types of SMEs. This study offers a new viewpoint on the heterogeneity from the standpoint of property rights and technological attributes, assisting scholars to better comprehend how SMEs' synergistic transformation of digitization and greenization affects enterprise risk-taking.

## Literature review

Enterprise risk-taking reflects a choice and decision made by an enterprise in the face of future income uncertainty. The external economic environment [18–20] and the internal enterprise environment [21–23] affect enterprise risk-taking. The development of the digital economy and green economy changes the development environment of enterprises. In order to adapt to the surrounding environment, enterprises actively carry out digital and green transformations, affecting the level of enterprise risk-taking.

The empirical evidence of the impact of enterprise digital transformation on enterprise risk-taking is increasingly abundant. Scholars generally determined that enterprise digital transformation has a significantly positive promotion effect on enterprise risk-taking [7,8,15,24,25]. Nonetheless, scholars hold different viewpoints on the internal mechanism of the impact of digital transformation on enterprise risk-taking. According to information asymmetry theory, enterprise digital transformation helps improve enterprises' operating flexibility and financing flexibility by improving internal and external information asymmetry [8,24]. Thus, it improves resource allocation efficiency and firm value [9], reducing managers' risk aversion behaviours [15], and then enhancing enterprise risk-taking. Based on hierarchy theory and enterprise strategy theory, enterprise digital transformation helps improve management ability, strategic deviation, and asset utilization ability [25], thus improving enterprise risk-taking. In the perspective of innovation theory, enterprise digital transformation improves innovation capability [8,9], thus enhancing enterprise risk-taking.

There is rarely direct evidence about the relationship between enterprise green transformation and risk-taking. However, enterprise green behavior has a positive effect on promoting risk-taking. According to agency theory, green credit policies promote enterprise risk-taking by reducing agency costs and easing financing constraints [10]. On the basis of information asymmetry theory, Zhang [26] established that enterprise risk-taking promotes the positive influence of green supply chain management on firm value. Li [12] found the fact that green mergers and acquisitions of enterprises alleviate the credit constraints enterprises face and reduce the tax burden, increasing enterprise risk-taking.

Scholars' research on the synergistic transformation of digitization and greenization focuses on the impact of digital transformation on green transformation. Zhang [1] discovered that digital transformation promotes enterprise green transformation by alleviating information asymmetry. More scholars focus on the relationship between enterprise digital transformation and green technology innovation. For example, enterprise digital transformation help alleviate financing constraints [2,5,27], attract government subsidies [3,5], increase R&D investment [5], and improve management efficiency [28], further enhance enterprise green technology innovation. Du [3] proposed that enterprise digital transformation alters enterprise factor allocation and energy production mode, promotes information transparency, reduces transaction costs, and helps promote energy technology innovation. On the other hand, Dou & Gao [4] pointed out that enterprise digital transformation helps promote green technology innovation. Further study fount that the impact is more significant for state-owned enterprises, manufacturing enterprises, and enterprises in regions with strict environmental supervision. Some scholars have also determined that greenization has a pulling effect on promoting the application of digital technology. For example, Yu [29] assessed that energy enterprises actively use digital technologies to improve their production processes and management systems to reduce pollutant emissions and avoid environmental violations. From the above analysis, it can be seen that digitalization and greenization have a mutually reinforcing effect. However, no scholars have measured the degree of synergistic transformation of digitization and greenization and analyzed its economic consequences. There is still a gap in this area of research.

Academics highly value the study of enterprise risk-taking, green transformation, and digital transformation since it helps us comprehend the financial effects of digitization and greenization. Firstly, enterprise digital transformation helps improve the efficiency of information transmission, organizational efficiency, and innovation ability, which promotes enterprise risk-taking. Secondly, enterprise green transformation helps reduce agency costs, alleviate information asymmetry, and then promote enterprise risk-taking. Thirdly, enterprise digital transformation helps promote green technology

innovation and green transformation. Green transformation also has a pulling effect on the application of digital technology. However, the following problems remain unsolved in the existing research: Firstly, the existing literature does not pay attention to the measurement of synergistic transformation of digitization and greenization. This paper uses the Haken model to measure the SMEs' synergistic transformation of digitization and greenization, further deepening the research in this field. Secondly, the existing literature analyzes the economic consequences of digital or green transformation from a single perspective. This study further explores the impact of SMEs' synergistic transformation of digitization and greenization on risk-taking. Thirdly, the mediating effect of cash holdings and R&D investment on the impact of SMEs' synergistic transformation of digitization and greenization on risk-taking is not examined. This paper attempts to answer the above questions, extend the application scope of the Haken model, and provide empirical evidence on the impact of SMEs' synergistic transformation of digitization and greenization on risk-taking.

## Research hypothesis

### SMEs synergistic transformation of digitization and greenization and enterprise risk taking

Haken proposed synergetics theory in 1971 [30]. Synergetics theory holds that two environmental subsystems have mutual influence and cooperation [31]. Lutfullaevich proposed that in sociological research synergetics is also an important theory for scholars to explain existing social problems [32]. Suppose an open system wants to achieve synergy. In that case, it needs to promote the fluctuation of the system through continuous exchange of matter and energy with the outside world and constantly receive positive feedback from the outside world to self-organize the transformation from disorder to order.

The development of digital technology has gradually changed the financial strategy of enterprises [33], technological upgrade [34,35], and organizational structure [36,37]. Digital technology can be applied in different systems at a very low cost. Digitization and greenization are important issues faced by SMEs today. The digital system and the green system both compete and cooperate. The digital system provides support and empowerment for the green system [38], and the green system provides promotion and traction for the digital system [29]. Digital and green systems have gradually evolved into synergistic systems of digitization and greenization, becoming a new path for SMEs to achieve win-win economic benefits and environmental protection.

Synergistic transformation of digitization and greenization enables SMEs to apply digital technologies to digital systems and green systems simultaneously and at a low cost, achieving cost savings and value addition through sharing, thus generating synergies. Firstly, synergistic transformation of digitization and greenization promotes the integration of technology and management, improves the quality, efficiency, and effect of green innovation, and the integration of the two further gives birth to new business forms and models [39]. The essence of digital transformation is to improve efficiency [40]. Digital technology provides resources and technologies for SMEs to synergize digitization and greenization and guarantee the transformation of green production technology. Green technology innovation further improve the financial performance of enterprises by changing the production mode [41] and enhancing the risk-taking capacity of SMEs. Secondly, a synergistic transformation of digitization and greenization helps SMEs build a unique competitive advantage. The severe environmental and climate crisis brought by the traditional mode of production has become an important factor restricting the sustainable development of a social economy. Hence, rational use of existing resources to achieve environmentally sustainable development has become a strong demand from suppliers, customers, society, and other stakeholders [42,43]. Synergistic transformation of digitization and greenization enables SMEs to conduct green innovation activities with more reasonable use of digital technologies, form a unique green production system, help enterprises win the favor of stakeholders, bring unique competitive advantages to enterprises [44], and meet the sustainable development of SMEs.

Based on the above analysis, SMEs' synergistic transformation of digitization and greenization is helpful to enterprise risk-taking. Therefore, hypothesis H1 is proposed:

H1: SMEs' synergistic transformation of digitization and greenization has a significant positive impact on enterprise risk-taking.

## The mediating effect of cash holdings

According to *the International Monetary Fund*, the digital sector impacts the enterprise's core activities [45]. In digital transformation, enterprises establish digital platforms to achieve fundamental business, operational, and organizational change [46]. Consequently, enterprises build large databases to enhance information transparency by hardening soft information [47] and accelerating the information exchange between internal and external enterprises [48].

Enterprise digital transformation also improves the speed and efficiency of external stakeholders' access to internal information [35]. This digital hardware and software information provides low-cost and high-efficiency infrastructure for SMEs' synergistic transformation of digitization and greenization. Firstly, SMEs' synergistic transformation of digitization and greenization is conducive to reducing the cash holdings by the agent motives and thus increasing enterprise risk-taking. The more serious the agency conflict, the higher the level of cash holdings based on agency motives [49,50]. Furthermore, the synergistic transformation of digitization and greenization helps SMEs realize the coordinated development of digital and green systems and promotes enterprises' use of digital resources to realize green technology innovation [5] and business model reconstruction [34]. These changes are based on a high degree of alignment between the goals of shareholders and managers. The consistency of the two objectives enables enterprises to concentrate superior resources to carry out related investment activities efficiently.

SMEs' synergistic transformation of digitization and greenization enables shareholders to obtain digital and green investment information more conveniently and effectively, implement low-cost and efficient incentives, and supervise the behavior of managers. When SMEs face better investment opportunities, managers' cost of pursuing private returns increases, and managers' motivation to hold excess cash decreases [50]. Cash holdings by enterprises are more inclined to implement digital and green investment projects that maximize shareholder wealth and increase enterprise risk-taking. SMEs' synergistic transformation of digitization and greenization helps reduce precautionary cash holdings and thus increase enterprise risk-taking. SMEs face a high degree of economic fluctuations and financing constraints. In order to avoid illiquidity, SMEs need to hold a large amount of cash to meet the need for precautionary motives.

SMEs' synergistic transformation of digitization and greenization provides good information transparency, which helps SMEs to provide creditors with more abundant and effective credit status information [51] and alleviates agency problems between shareholders and creditors. SMEs need to obtain financial support for digital and green investment projects without holding large amounts of cash, which reduces cash holdings for precautionary motives. Strong financing capacity helps to improve the level of digital and green investment of SMEs, which in turn increases enterprise risk-taking. It can be seen that SMEs' synergistic transformation of digitization and greenization helps reduce the cash holdings with agency and precautionary motives, thus increasing enterprise risk-taking. Therefore, hypothesis H2 is proposed:

H2: SMEs' synergistic transformation of digitization and greenization increases enterprise risk-taking by reducing cash holdings.

## The mediating effect of R&D investment

To achieve sustainable development, enterprises must comply with environmental control, reduce the negative impact of products or services on the environment, and pay attention to the environmental requirements of stakeholders. These requirements necessitate radical improvements in products and services in existing target markets using new processes. Synergistic transformation of digitization and greenization is conducive to integrating SMEs, expanding the depth and breadth of digital knowledge and green knowledge, performing R&D activities to improve production processes, and minimizing waste emissions from the source. It helps enterprises prevent ecological pollution, improve environmental performance [52], and reduce the cost of environmental regulation [53]. However, the limited nature of resources determines

that the R&D funds of SMEs are relatively limited. Investing in digital or green systems alone leads to neglect of one or the other, which is not conducive to improving resource efficiency. This situation has led SMEs to constantly seek two-way collaboration between digital and green R&D investment to maximize resource utilization efficiency.

Synergistics theory points out that if an open system wants to achieve synergy, it must continuously exchange matter and energy with the outside world. It promotes the realization of system fluctuations and constantly receives positive feedback from the outside world to self-organize the transformation from disorder to order. In the synergistic system of digitization and greenization in SEMs, fluctuations are reflected in the integration and innovation of digital technology and green technology within and between enterprises to achieve green technology innovation. The higher the degree of SMEs' synergistic transformation of digitization and greenization, the higher the collaboration between the digital and green subsystems. Digital and green knowledge and technology interact in a certain way to achieve sharing and public use, coordinate and cooperate with R&D activities, and improve resource allocation efficiency. Note that effective resource utilization helps enterprises reduce costs [54]. It can reduce R&D investment without reducing innovation output, improve the efficiency of R&D investment, and thus increasing enterprise risk-taking. Therefore, hypothesis H3 is proposed:

H3: SMEs' synergistic transformation of digitization and greenization increases enterprise risk-taking by reducing R&D investment.

Fig 1 illustrates the mechanism of the SMEs' synergistic transformation of digitization and greenization on risk-taking. First of all, the system of digitization and greenization in SMEs promotes integrating technology and management, helping SMEs establish unique competitive advantages. These are powerful factors that promote enterprise risk-taking. Secondly, SMEs' synergistic transformation of digitization and greenization improves the synchronous operation among internal organizations, reduces the cash holding by agency motivation and prevention motivation, and helps promote enterprise investment expenditure and risk-taking. Thirdly, the system of digitization and greenization in SMEs helps to improve the

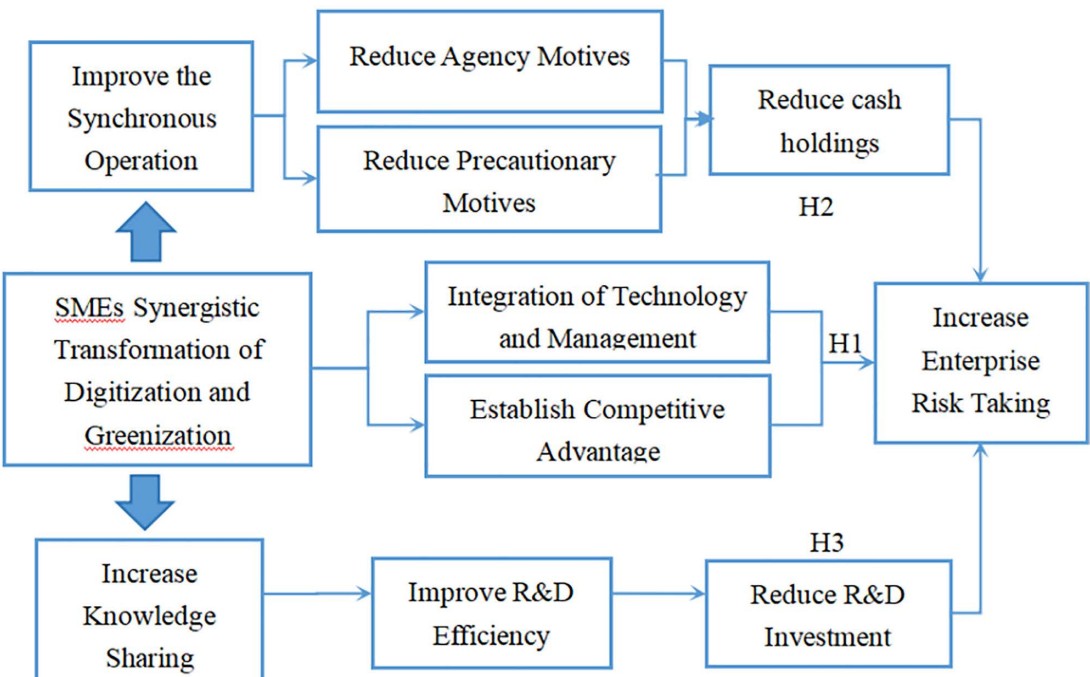

**Fig 1. The impact of SMEs' synergistic transformation of digitization and greenization on enterprise risk-taking.**

sharing of knowledge, reduce the investment scale of innovation investment per unit of output, improve the efficiency of R&D investment, and thus increasing risk-taking.

## Research design

### Data resource

This study analyzes the data of enterprises listed on SME boards from 2012 to 2022 as samples, taking into account the effects of the 2008 financial crisis on the capital market and the requirement for measuring risk-taking. This study obtained research data from the China Stock Market & Accounting Research Database (CSMAR). The data of annual reports published by listed SMEs adopted the following screening criteria: Firstly, due to the different operating conditions of financial and insurance companies, this paper excluded such companies. Secondly, ST, *ST, and PT were excluded from this paper due to the significant decline in firm performance and the government's implementation of special regulatory measures. Thirdly, the calculation of enterprise risk-taking requires at least 3 years of data. This paper excluded companies listed for less than 3 years or continuously disclosed profitability data for less than 3 years. Fourthly, the data of insolvency is excluded in this paper. In order to preserve the data as much as possible, unbalanced panel data was used in this paper. Finally, this paper obtains 7410 research data from 889 SMEs. There are missing data in cash holdings and R&D investment, with a sample size of 6995 and 6741, respectively. In order to prevent the outliers from influencing the regression results, the continuous variables were winsorized (double-tailed) at 5% quantile. The mining of the annual report text was completed using Python 3.10. Excel 2016 and Stata 16.0 were used to complete the initial data screening and processing of samples.

### Models

The rationality of model selection affects the accuracy of research results, so the following tests are conducted in this paper: First, the F-test is performed. The results show that Prob > F = 0.0000, indicating that the fixed effect model is more suitable. Secondly, the Hausman test is conducted. The results show that Prob > chi2 = 0.0000, indicating that the original hypothesis is rejected and the fixed effect results are more suitable. Combining the F-test and Hausman test, the fixed effect model is selected.

Model (1) tests the impact of SMEs' synergistic transformation of digitization and greenization on enterprise risk-taking.

$$
\begin{aligned}
RISK_{itj} = &\alpha_0 + \beta_1 \times SDG_{itj} + \beta_2 \times ROE_{itj} + \beta_3 \times AGE_{itj} + \beta_4 \times DEBT_{itj} + \beta_5 \times BSIZE_{itj} \\
&+ \beta_6 \times INTEN_{itj} + \beta_7 \times TAT_{itj} + \beta_8 \times SHARE_{itj} + \beta_9 \times DUAL_{itj} + \theta + \vartheta + \varepsilon
\end{aligned}
\tag{1}
$$

Referring to Wen and Ye [55], the sequential regression method examines the mediating effect of cash holdings and R&D investment in the impact of SMEs' synergistic transformation of digitization and greenization on enterprise risk-taking.

$$
\begin{aligned}
MED_{itj} = &\alpha_0 + \beta_1 \times SDG_{itj} + \beta_2 \times ROE_{itj} + \beta_3 \times AGE_{itj} + \beta_4 \times DEBT_{itj} + \beta_5 \times BSIZE_{itj} \\
&+ \beta_6 \times INTEN_{itj} + \beta_7 \times TAT_{itj} + \beta_8 \times SHARE_{itj} + \beta_9 \times DUAL_{itj} + \theta + \vartheta + \varepsilon
\end{aligned}
\tag{2}
$$

$$
\begin{aligned}
RISK_{itj} = &\alpha_0 + \beta_1 \times SDG_{itj} + \beta_2 \times MED_{itj} + \beta_3 \times ROE_{itj} + \beta_4 \times AGE_{itj} + \beta_5 \times DEBT_{itj} \\
&+ \beta_6 \times BSIZE_{itj} + \beta_7 \times INTEN_{itj} + \beta_8 \times TAT_{itj} + \beta_9 \times SHARE_{itj} + \beta_{10} \times DUAL_{itj} \\
&+ \theta + \vartheta + \varepsilon
\end{aligned}
\tag{3}
$$

where $MED_{itg}$ is the mediating variable, α is the intercept term, $\beta_1$, $\beta_2$, ……$\beta_{10}$ is the coefficient to be estimated, θ is the annual fixed effect term, $\vartheta$ is the industry fixed effect term, and $\in$ is the residual term.

## Variables

**Dependent variable.** In existing studies, scholars mostly use financial performance to measure enterprise risk-taking [56,57]. This paper references Tian [8] and Dai [9], which use the standard deviation of the ratio of the enterprise's income before interest, tax, and amortization to assets in three years to measure enterprise risk-taking. The measurement is as follows:

$$E_{i,c,t} = \frac{EBITDA_{i,c,t}}{A_{i,c,t}} - \frac{1}{N} \sum_{K=1}^{N_{i,c,t}} \frac{EBITDA_{k,c,t}}{A_{k,c,t}} \tag{4}$$

$$RISKt = \sqrt{\frac{1}{T-1} \sum_{t=1}^{T} \left( E_{i,c,t} - \frac{1}{T} \sum_{t=1}^{T} E_{i,c,t} \right)} \quad T = 3 \tag{5}$$

where $\frac{EBITDA}{A}$ is the ratio of an enterprise's income before interest, tax and amortization to assets.

**Independent variables.** Synergetics is the study of the process from imbalance to equilibrium in the development of affairs and has broad application prospects in social science [32]. After the birth of synergetics, many scholars in social science research use this theory to explain the synergistic relationship between subsystems in an enterprise. For example, Guasmin & Rajindra [58] analyzed the synergistic action between human resource management, strategic marketing management, and financial performance. According to Synergetics, the synergistic transformation of digitization and greenization is an open, non-equilibrium system. The system has evolved from disorder to order since its emergence. Therefore, it is necessary to establish an order parameter equation to identify the degree of synergistic transformation of digitization and greenization of each enterprise. The Haken model describes how the subsystems of complex systems spontaneously form macroscopic ordered structures through nonlinear interactions and synergies. Without external intervention, the system can automatically evolve into an orderly structure or function according to specific laws. There are fast variables and slow variables in the system. Slow variables dominate fast variables, and the system's long-term behavior depends on a few order parameters. Thus, slow variables should be found to determine the synergistic effect of two subsystems, and order parameter equations should be established. Nonlinear dynamics equations usually describe the Haken model. Its general form is:

$$\dot{q}(t) = N(q, \nabla, \alpha) + F(t) \tag{6}$$

where $q$ is a variable that describes the system's state; $N$ is a nonlinear function reflecting the interaction between subsystems; $\alpha$ is the control parameter that determines the system's stability; $F(t)$ is a random fluctuating force. The Haken model is widely used in social science research. For example, Yang [13] selected five big cities in China as samples to analyze the dynamic co-evolution law of economy and logistics. It was found that metropolitan economic growth is the order parameter that promotes the coordinated development of the metropolitan economy and logistics. Zhong [14] analyzed and predicted the synergistic between economic growth and energy consumption in the Beijing-Tianjin-Hebei region. Du & Yang [59] analyzed the co-evolution law of the logistics industry and manufacturing industry in the Yangtze River Economic Belt, and discovered that the logistics industry plays an increasingly important role in the relationship between them. Based on the above research, this paper uses the Haken model to measure the degree of synergistic transformation of digitization and greenization. The calculation process is described as follows:

The synergistic system of digitization and greenization is an open symbiotic system, including two digital and green subsystem components. The state variables of the two subsystems are represented by $q_1$ and $q_2$, respectively. Here, $q_1$

is the driving system evolution subsystem (slow variable) and $q_2$ is the servo subsystem (fast variable). Consequently, the evolution equation of the synergistic system of digitization and greenization is as follows:

$$\dot{q}_1 = -\gamma_1 q_1 - aq_1 q_2 \tag{7}$$

$$\dot{q}_2 = -\gamma_2 q_2 + bq_1{}^2 \tag{8}$$

where $\dot{q}_1$ and $\dot{q}_2$ represent the derivative function of the state variable concerning time, $\gamma_1$ and $\gamma_2$ represent the damping coefficient, and $a$ and $b$ represent the strength of the interaction of the state variable. The parameters' size $\gamma_1$, $\gamma_2$, $a$, and $b$, reflect the evolution behavior of the synergistic system of digitization and greenization. When the system reaches $q_1 = q_2 = 0$, if $|\gamma_2| \geq |\gamma_1|$, and $\gamma_2 > 0$, then the adiabatic approximation hypothesis of the system is satisfied. At this time, $q_2$ is a fast parameter that decays rapidly and $q_1$ is a slow variable that plays a leading role in the system's evolution. Let $\dot{q}_2 = 0$ in equation (8), which yields:

$$q_2 \approx \frac{b}{\gamma_2} q_1{}^2 \tag{9}$$

Equation (8) reveals the domination-servo behavior between the digital and green subsystems. By substituting equation (9) into equation (7), the evolution equation of the order parameter is obtained:

$$\dot{q}_1 = -\gamma_1 q_1 - \frac{ab}{\gamma_2} q_1{}^3 \tag{10}$$

Equation (10) presents that $q_2$ varies with $q_1$. Thus, $q_1$ is the order parameter of the system (that is, the slow variable), and the potential function of the system can be obtained by inverse and integral processing:

$$V = \frac{1}{2}\gamma_1 q_1{}^2 + \frac{ab}{4\gamma_2} q_1{}^4 \tag{11}$$

Set the left side of the equation (11) equal to zero, and solve the three solutions of the potential function. That is: $q_1' = 0$, $q_1'' = \sqrt{\left|\frac{\gamma_1\gamma_2}{ab}\right|}$, $q_1''' = -\sqrt{\left|\frac{\gamma_1\gamma_2}{ab}\right|}$.

Since the zero solution of equation (11) is unstable, it is generally not considered in practical analysis. In contrast, the non-zero solution is stable, and the composite system forms a new stable state through mutation. Since annual data are used in this paper, the Haken model needs to be discretized:

$$q_1(t) = (1 - \gamma_1)\, q_1(t-1) - aq_1(t-1)\, q_2(t-1) \tag{12}$$

$$q_2(t) = (1 - \gamma_2)\, q_2(t-1) + bq_1{}^2(t-1) \tag{13}$$

To measure SMEs' synergistic system of digitization and greenization, this paper calculates enterprise digital transformation and green transformation and then puts them into the Haken model.

Based on the studies of Wang [60] and Tian [8], this paper uses text analysis to construct enterprise digital transformation. The first step is to set up a keyword library, which includes five dimensions of artificial intelligence technology: blockchain technology, cloud computing technology, big data technology, and digital technology application. Correspondingly,

197 keywords obtained. The second step is to count the keyword frequency. This paper uses the *Jieba* word segmentation function in *Python* to conduct a keyword search in the annual report published by enterprises listed on the SME board. Subsequently, count how often each keyword appears in the annual report (excluding the negative words before the keywords). The third step is to measure enterprise digital transformation using the natural logarithm of digital transformation word frequency.

According to the research of Guo [61], this paper also uses text analysis to construct enterprise green transformation. The first step is to set up the keyword library from the system, as well as the implementation and guarantee perspectives. Through manual retrieval, this paper initially obtains keywords from national documents related to green economy and low-carbon environmental protection. A complete keyword library is formed using machine learning. Consequently, 113 keywords are obtained. In the second step, this paper uses the *Jieba* word segmentation function in *Python* to search and match keywords in the enterprises listed on the SME board annual report. This paper calculates the frequency of each keyword in the annual report (excluding the words with negative expressions before the keywords). Finally, the natural logarithm of word frequency of green transformation is used to measure enterprise green transformation.

In this paper, enterprise digital transformation and green transformation are taken as order parameters and tested according to equations (12) and (13). Taking enterprise digital transformation as an order parameter, it is obtained that $\gamma_1 = 0.0727$, $\gamma_2 = 0.4036$, a=0.0086, b=0.0101, which satisfy the adiabatic approximation hypothesis. Taking enterprise green transformation as the order parameter, it is obtained that $\gamma_1 = 0.4237$, $\gamma_2 = 0.1022$, a=−0.01334, b=−0.0003, which does not satisfy the adiabatic approximation hypothesis. Therefore, enterprise digital transformation should be considered as an order parameter. By bringing $\gamma_1 = 0.0727$, $\gamma_2 = 0.4036$, a=0.0086, b=0.0101 into the evolution equation, we can obtain:

$$\dot{q}_1 = -0.0727q_1 - \frac{0.0086 \times 0.0101}{0.4036}q_1{}^3$$

(14)

The solution is $q_1' = 0$, $q_1'' = 18.3704$, and $q_1''' = -18.3704$. Then, the potential function of the composite system is obtained as follows:

$$V = 0.03635q_1{}^2 + 0.00005386q_1{}^4$$

(15)

Because enterprise digital transformation and green transformation are positive numbers, only the part of the potential function q>0 is considered. Taking $q_1'' = 18.3704$ into the potential function and the stable points of enterprise digital and green transformation (18.3704, 18.4006). In order to evaluate SMEs synergistic transformation of digitization and greenization, the *Euclidean Distance* is used to measure the degree of synergy (SDG):

$$SDG = \sqrt{(q - 18.3704)^2 + [V(q) - 18.4006]^2}$$

(16)

The larger the value of SDG, the farther away from the stable point, and the higher the degree of SMEs synergistic transformation of digitization and greenization.

**Mediating variable.** Cash holdings (CASH): Based on the studies of Gao [50], Amess [62], Demir and Ersan [63], this paper uses the ratio of cash assets to total assets to measure cash holdings. In order to better show empirical results, that ratio is multiplied by 100.

Research and Development (R&D) investment: According to the studies of Aw [64] and Li [65], this paper uses the natural logarithm of R&D investment amount to measure R&D investment.

**Control variables.** According to studies of John [56], Wu [51], Zhou [66], and Dai [9], this paper adopts Return on Equity (ROE), firm age (AGE), leverage (DEBT), board size (BSIZE), capital intensity (INTEN), total assets turnover (TAT),

concentration of shareholding (SHARE), CEO duality (DUAL) as control variables. The variable design of this paper is shown in Table 1.

## Empirical results and analysis

### Descriptive Statistics

The descriptive statistical characteristics of the main variables selected in this paper are displayed in Table 2. The data show that the average risk-taking (RISK) of enterprises listed on the SME board from 2012 to 2022 is 0.0484. According to the research by Deng [10], the average risk-taking of listed companies in the whole sectors from 2012 to 2020 is 0.0269. It can be seen that the risk-taking of enterprises listed on the SME board is higher than the overall level of listed companies,

**Table 1. Variables.**

| Variable Type | Variable Name | Variable Symbols | Variable Definition |
|---|---|---|---|
| Dependent Variables | Corporate risk taking | RISK | Standard deviation of EBITDA |
| Independent Variables | SMEs synergistic transformation of digitization and greenization | SDG | Calculated using the Haken model |
| Mediating variables | Cash holdings | CASH | $\frac{Closing balance of cash assets}{Closing balance of assets} \times 100$ |
| | R&D investment | RD | Ln(investment expense +1) |
| Control Variables | Return on equity | ROE | $\frac{NetProfit}{NetAssets}$ |
| | Firm age | AGE | Ln(Fiscal year – Establishment year+1) |
| | Leverage | DEBT | Total liabilities at end of period/total long assets at end of period |
| | Board size | BSIZE | Ln(Number of directors+1) |
| | Capital intensity | INTEN | $\frac{TotalAssets}{OperatingIncome}$ |
| | Total assets turnover | TAT | Current operating income/closing balance of assets |
| | Concentration of shareholding | SHARE | $\frac{NumberofSharesHeldbyTopTenShareholders}{TotalNumberofCompanyshares}$ |
| | CEO duality | DUAL | The dual roles of chairman and general manager are 1, otherwise 0 |

**Table 2. Descriptive statistics.**

| Variable | Obs | Mean | Std. Dev. | Min | Max |
|---|---|---|---|---|---|
| RISK | 7410 | .0419 | .0335 | .0067 | .1369 |
| SDG | 7410 | 24.8253 | 1.0994 | 20.8566 | 25.9236 |
| CASH | 6995 | 20.1786 | 1.1670 | 13.719 | 25.0013 |
| RD | 6741 | 17.9334 | 1.8303 | 0 | 23.7301 |
| ROE | 7410 | .0603 | .0799 | −.1517 | .1999 |
| AGE | 7410 | 2.0114 | .4921 | .6931 | 3.0445 |
| DEBT | 7410 | .1029 | .1122 | .0010 | .3761 |
| BSIZE | 7410 | 2.2948 | .2067 | 0 | 3.091 |
| INTEN | 7410 | 2.1284 | 1.1499 | .7475 | 5.1116 |
| TAT | 7410 | .6400 | .4722 | .0034 | 7.6092 |
| SHARE | 7410 | 57.7939 | 14.2187 | 15.4403 | 95.9978 |
| DUAL | 7410 | .3274 | .4693 | 0 | 1 |

which is in line with the characteristics of high growth of SMEs. The maximum value of SMEs' synergistic transformation of digitization and greenization (SDG) is 25.9236, the minimum value is 20.8566, and the average value is 24.8253. It can be seen that the degree of SMEs' synergistic transformation of digitization and greenization is relatively high.

## Correlation analysis

The correlation between SMEs' synergistic transformation of digitization and greenization (SDG) and enterprise risk-taking (RISK) is 0.0255, which is significant at 5% level. It indicates a significant positive correlation between SMEs' synergistic transformation of digitization and greenization and enterprise risk-taking, which preliminary supports hypothesis H1. Cash holdings are negatively correlated with SMEs' synergistic transformation of digitization and greenization and enterprise risk-taking, and R&D investment (RD) is negatively correlated with SMEs' synergistic transformation of digitization and greenization and enterprise risk-taking, which are significant at 1% level. Hypothesis H2 and H3 are preliminarily supported. The correlation coefficients between the selected control variables and enterprise risk-taking are all significant at 1% level. In contrast, the correlation between control variables is weak. The Variance Inflation Factor (VIF) of the selected variables (SDG, ROE, AGE, DEBT, BSIZE, INTEN, TAT, SHARE, DUAL) is 1.06, 1.01, 1.24, 1.06, 1.02, 1.08, 1.10, 1.17, 1.04 respectively, and the average value of 1.09, indicating that there is no high col-linearity between the variables.

## Regression analysis

**SMEs synergistic transformation of digitization and greenization and enterprise risk-taking.** Column (1) displayed in Table 3 presents that the impact of SMEs' synergistic transformation of digitization and greenization (SDG) on enterprise risk-taking (RISK) is 0.0067, which is significant at 1% level, indicating that SMEs' synergistic transformation of digitization and greenization has a significant promotion effect on enterprise risk-taking, which supports hypothesis H1. In the development of digital economy and green economy, enterprise digital transformation is conducive to improving organizational efficiency [40], promoting green innovation [3], and establishing unique competitive advantages. Changing the production mode of enterprises can further improve financial performance [41] and enterprise risk-taking [8,9]. Zhang [1] determined that enterprises need abundant data support for green transformation, which has a reverse promotion effect on SMEs' digital transformation, promoting SMEs to perform digital investment and changing how SMEs profit acquisition. Enterprise digital transformation and green transformation are conducive to the development of SMEs. SMEs' synergistic transformation of digitization and greenization can reduce the investment cost of SMEs' digital transformation and green transformation through the sharing mechanism. Consequently, it also improves production efficiency through green technology innovation and financial performance and promotes enterprise risk-taking.

**The mediating effect of cash holdings.** Column (2) displayed in Table 3 shows that the influence coefficient of SMEs' synergistic transformation of digitization and greenization (SDG) on cash holdings (CASH) is −0.1049, which is significant at 1% level, indicating that SMEs' synergistic transformation of digitization and greenization can significantly reduce cash holdings. Column (3) displayed in Table 3 presents that the impact of cash holdings on enterprise risk-taking is −0.0168, which is significant at a 1% level. This indicates that cash holdings mediate the impact of SMEs' synergistic transformation of digitization and greenization on enterprise risk-taking, which supports hypothesis H2. According to agency theory, aligning the goals of shareholders and management helps enterprises concentrate superior resources to implement low-cost and efficient investment activities and reduce the cash holdings by management for pursuing private gains [62]. The SMEs' synergistic transformation of digitization and greenization helps to reduce the agency cost of enterprises and then reduces the cash holdings of SMEs. It improves the investment efficiency of digitization and greenization and enhances enterprise risk-taking. Based on information asymmetry theory, good information transparency helps alleviate agency problems between shareholders and creditors and reduce the cash holdings by SMEs due to precautionary motives. SMEs' synergistic transformation of digitization and greenization provides the outside world with more abundant and

**Table 3. Regression result.**

| | (1) RISK | (2) CASH | (3) RISK | (4) RD | (5) RISK |
|---|---|---|---|---|---|
| SDG | .0067*** (3.0183) | −.1049*** (−6.7817) | .0051** (2.2475) | −.0719** (−2.2637) | .0056** (2.3784) |
| CASH | | | −.0168*** (−8.8976) | | |
| RD | | | | | −.0035*** (−3.5538) |
| ROE | −.0044*** (−11.2637) | .0055** (2.089) | −.0043*** (−11.066) | .0027 (.505) | −.0042*** (−10.6901) |
| AGE | −.0276*** (−2.6643) | −.1928** (−2.4364) | −.0283** (−2.4384) | −.246 (−1.497) | −.0326*** (−2.676) |
| DEBT | .1389*** (17.0943) | −.1772*** (−3.0737) | .1546*** (18.2566) | .4494*** (3.5954) | .1823*** (19.6415) |
| BSIZE | −.0152** (−2.0648) | .0465 (.908) | −.0152** (−2.018) | .2743*** (2.6011) | −.0138* (−1.7611) |
| INTEN | .0008** (2.3902) | .0011 (.4978) | .0007** (2.2981) | −.0062 (−1.3763) | .0006* (1.8335) |
| TAT | .018*** (3.8028) | −.1635*** (−4.8488) | .0104** (2.1065) | .0540 (.7764) | .0133*** (2.5816) |
| SHARE | −.0005** (−2.5435) | .0161*** (12.6515) | −.0003 (−1.4848) | .0069*** (2.6533) | −.0005*** (−2.6828) |
| DUAL | .0088** (2.3831) | .107*** (4.1371) | .0107*** (2.8037) | .1085** (2.0511) | .0087** (2.2241) |
| constant term | 0 (−.0002) | 23.1001*** (42.074) | .3806*** (4.1561) | 17.1425*** (14.416) | .1125 (1.2531) |
| Observations | 7410 | 6995 | 6995 | 6741 | 6741 |
| R2 | .1008 | .3081 | .1240 | .1807 | .1200 |

*** $p < .01$, ** $p < .05$, * $p < .1$.

effective information, attracts more effective financial support for digital and green investment, reduces cash holdings due to precautionary motives, improves enterprise investment efficiency, and enhances enterprise risk-taking.

**The mediating effect of R&D investment.** Column (4) displayed in Table 3 shows that the influence coefficient of SMEs' synergistic transformation of digitization and greenization (SDG) on R&D investment (RD) is −0.0719. This is significant at 5% level, indicating that SMEs' synergistic transformation of digitization and greenization can significantly reduce R&D investment. Column (5) displayed in Table 3 shows that the impact of R&D investment on enterprise risk-taking is −0.0035, which is significant at a 1% level. The results examines how R&D investment mediates the impact of SMEs' synergistic transformation of digitization and greenization on enterprise risk-taking, which supports hypothesis H2. According to the synergetics theory, SMEs' synergistic transformation of digitization and greenization can maximize the effect of digital knowledge and green knowledge through competition and cooperation and promote knowledge spillover and new knowledge creation. Enterprises continue to promote digital transformation in the context of digital and green economies [34,65]. Hence, integrating digital and green knowledge helps enterprises concentrate superior resources to conduct innovation activities and constantly seek coordination and cooperation in digital and green innovation investment. Concentrated investment in innovation helps reduce the cost of enterprise R&D, thus enhancing enterprise risk-taking.

## Heterogeneity analysis

**Property Rights.** According to previous studies, the impact of enterprise digital transformation on risk-taking is influenced by the nature of property rights [8,9]. Analyzing the impact of SMEs' synergistic transformation of digitization and greenization on enterprise risk-taking from the property rights perspective helps promote SMEs' synergistic transformation according to ownership characteristics (Table 4).

Columns (1) – (2) in Table 4 show that the influence coefficient of synergistic transformation of digitization and greenization of non-state-owned SMEs on enterprise risk-taking is 0.0080, which is significant at 1% level. The influence coefficient of synergistic transformation of digitization and greenization of state-owned SMEs is −0.0044, which is significant at 10% level. It can be seen that the synergistic transformation of digitization and greenization of non-state-owned SMEs has a positive impact on enterprise risk-taking. In contrast, that of state-owned SMEs has a negative impact. Digital transformation means a comprehensive transformation of enterprise strategy [33] and organizational structure [37], which improves resource utilization efficiency. Ferguson [67] assessed that the essence of enterprise green transformation is to improve resource utilization efficiency. Therefore, the synergistic transformation of digitization and greenization helps to improve operational efficiency for sustainable development. Compared with state-owned SMEs, non-state-owned SMEs

**Table 4. Heterogeneity analysis from Property Rights.**

| | Main effect | | The mediating effect of cash holdings | | | | The mediating effect of R&D investment | | | |
|---|---|---|---|---|---|---|---|---|---|---|
| | (1) Non state-owned enterprises | (2) State-owned enterprises | (3) Non state-owned enterprises | | (4) State-owned enterprises | | (5) Non state-owned enterprises | | (6) State-owned enterprises | |
| | RISK | RISK | CASH | RISK | CASH | RISK | RD | RISK | RD | RISK |
| SDG | .0080*** (3.1007) | −.0044* (−1.9317) | −.1129*** (−6.6938) | .0061** (2.278) | −.0754* (−1.9209) | −.0050** (−2.1654) | −.1068*** (−3.2842) | .0063** (2.3091) | .2565** (2.1989) | −.0030 (−1.2138) |
| CASH | | | | −.0184*** (−8.264) | | −.0012 (−.6599) | | | | |
| RD | | | | | | | | −.0040*** (−3.3568) | | .0007 (1.0345) |
| ROE | −.0043*** (−10.2913) | −.0014 (−.1412) | .0049* (1.8488) | −.0042*** (−10.039) | 1.0729*** (6.1228) | −.0017 (−.1644) | .0023 (.4429) | −.0041*** (−9.7409) | .0551 (.1108) | −.0031 (−.2942) |
| AGE | −.0358*** (−2.8991) | −.0134 (−1.277) | −.2374*** (−2.6762) | −.0362*** (−2.5984) | −.1704 (−.8688) | −.0092 (−.7989) | −.1798 (−1.0434) | −.0393*** (−2.7354) | −.1646 (−.2796) | −.0091 (−.7261) |
| DEBT | .1561*** (16.8398) | −.0191* (−1.6479) | −.2286*** (−3.7183) | .1719*** (17.7809) | .3616* (1.7698) | −.0242** (−2.0218) | .399*** (3.1677) | .2041*** (19.4085) | .9682 (1.6202) | −.0309** (−2.4156) |
| BSIZE | −.0211** (−2.3708) | −.0072 (−1.1693) | .0157 (.2694) | −.0221** (−2.4235) | .3089*** (2.8669) | −.0084 (−1.3299) | .3804*** (3.3831) | −.0207** (−2.2017) | −.0094 (−.0297) | −.0066 (−.9671) |
| INTEN | .0007** (2.0168) | .0012 (1.1326) | .0014 (.6303) | .0007** (1.9743) | .039* (1.69) | .0004 (.2991) | −.0057 (−1.3318) | .0005 (1.5095) | −.1134 (−1.0517) | .0004 (.1647) |
| TAT | .0202*** (3.7261) | −.0026 (−.4434) | −.1456*** (−4.0062) | .012** (2.1027) | −.4356*** (−4.1046) | −.0061 (−.9747) | .0801 (1.1364) | .0145** (2.4643) | −.5043 (−1.4682) | −.0051 (−.6921) |
| SHARE | −.0005** (−2.4092) | −.0003* (−1.7359) | .0164*** (11.6544) | −.0003 (−1.425) | .0084*** (2.6234) | −.0003 (−1.6422) | .008*** (2.9352) | −.0006*** (−2.6353) | −.0099 (−1.0506) | −.0003 (−1.5428) |
| DUAL | .0108** (2.5126) | −.0034 (−.9017) | .0962*** (3.4025) | .0125*** (2.8016) | .1932*** (2.9633) | −.0028 (−.7269) | .0918* (1.6876) | .0107** (2.3487) | .0935 (.4926) | −.005 (−1.2374) |
| constant term | −.0816 (−.9081) | .2938*** (4.1926) | 23.7757*** (38.9972) | .3242*** (2.9631) | 23.2269*** (18.1995) | .267*** (3.0928) | 18.3155*** (14.0087) | −.0303 (−.272) | 13.9706*** (3.6994) | .1800** (2.2151) |
| Observations | 6157 | 1253 | 5799 | 5799 | 1196 | 1196 | 5663 | 5663 | 1078 | 1078 |
| R2 | .1087 | .1072 | .2999 | .1332 | .4289 | .1091 | .1833 | .1298 | .198 | .1221 |

*** p < .01, ** p < .05, * p < .1.

face greater competitive pressure and stronger resource constraints. In order to effectively utilize existing resources to establish new competitive advantages, non-state-owned SMEs must concentrate on superior resources to improve organizational efficiency [68], which enhances investment efficiency and thus increasing enterprise risk-taking.

Column (3) – (4), displayed in Table 4, show that the influence coefficient of synergistic transformation of digitization and greenization of non-state-owned SMEs on cash holdings is −0.1129, and the impact of cash holdings on enterprise risk-taking is −0.0184, both of which are significant at 1% level. It shows that cash holdings mediate the impact of the synergistic transformation of digitization and greenization of non-state-owned SMEs on enterprise risk-taking. The influence coefficient of synergistic transformation of digitization and greenization of state-owned SMEs on cash holdings is −0.0754, which is significant at 1% level. The influence of cash holdings on enterprise risk-taking is −0.0012, which is insignificant. Non-state-owned enterprises are more constrained by resources than state-owned enterprises. They are under more significant pressure from competition. In the synergistic transformation of digitization and greenization, it is easier for non-state-owned SMEs to have a unified understanding, concentrate superior resources to carry out digital and green investments, reduce cash holdings, and thus improving enterprise risk-bearing.

Columns (5) – (6), displayed in Table 4, show that the influence coefficient of synergistic transformation of digitization and greenization of non-state-owned SMEs on R&D investment is −0.1068. The impact of R&D investment on enterprise risk-taking is −0.00404, both significant at 1% level. It indicates that R&D investment mediates the impact of the synergistic transformation of digitization and greenization of non-state-owned SMEs on enterprise risk-taking. The influence coefficient of synergistic transformation of digitization and greenization of state-owned SMEs on R&D investment is 0.2565, which is significant at a 5% level. The influence of R&D investment on enterprise risk-taking is 0.0007. However, this influence is insignificant.

The main reason is that non-state-owned enterprises are smaller and easier to adjust their strategies than state-owned enterprises. These enterprises establish digital transformation organizations, mobilize their resources to build digital platforms, and achieve synergistic transformation of digitization and greenization through green investment, green patent research, and development. Enterprise digital transformation supports and empowers green transformation [38], and enterprise green transformation promotes and draws digital transformation. Synergistic transformation of digitization and greenization of non-state-owned SMEs promotes resource allocation optimization. It enhances the integration of digitalization and greenization, reduces R&D investment, and thus increasing enterprise risk-taking.

**Technological attributes.** From the perspective of technological attributes, this paper analyzes the impact of SMEs' synergistic transformation of digitization and greenization on enterprise risk-taking. The results are shown in Table 5.

Columns (1) – (2) in Table 5 present that the impact of the synergistic transformation of digitization and greenization on enterprise risk-taking of non-high-tech SMEs is 0.0123, which is significant at 5% level. The impact of high-tech SMEs is 0.0001, but not significant. The possible reason is that high-tech SMEs have a high level of science and technology and advantages in R&D investment in their main business areas. In order to maintain the image of high-tech enterprises, the investment in their primary business needs to be maintained, and their investment scale in the synergistic transformation of digitization and greenization is limited. Non-high-tech SMEs can concentrate their advantageous resources to carry out exceptional investment in the synergistic transformation of digitization and greenization, improve investment efficiency, and enhance enterprise risk-bearing.

Column (3), displayed in Table 5, shows that the impact of the synergistic transformation of digitization and greenization on cash holdings of non-high-tech SMEs is −0.1229. The impact of cash holdings on enterprise risk-taking is −0.0328, both of which are significant at 1% level, indicating that the synergistic transformation of digitization and greenization of non-high-tech SMEs improves enterprise risk-taking by reducing cash holdings. The main reason is that in non-high-tech SMEs, the digital model effectively supports business and cross-organizational innovation, improves the degree of synergistic transformation of digitization and greenization, reduces the cash holdings due to inefficient investment, and thus increasing enterprise risk-taking.

Table 5. Heterogeneity analysis from Technological attributes.

| | Main effect | | The mediating effect of cash holdings | | The mediating effect of R&D investment | |
| --- | --- | --- | --- | --- | --- | --- |
| | (1) Non-high-tech | (2) High-tech | (3) Non-high-tech | | (4) Non-high-tech | |
| | RISK | RISK | CASH | RISK | RD | RISK |
| SDG | .0123** (2.2804) | .0001 (.1006) | −.1229*** (−5.017) | .0082 (1.4699) | −.0073 (−.1088) | .0107* (1.667) |
| CASH | | | | −.0328*** (−6.8701) | | |
| RD | | | | | | −.0059*** (−2.7659) |
| ROE | −.0029*** (−4.4383) | −.0171*** (−11.2404) | .0036 (1.2435) | −.0027*** (−4.1335) | −.0009 (−.1299) | −.0027*** (−3.8423) |
| AGE | −.0669*** (−2.7041) | −.0025 (−.3594) | −.2048* (−1.6598) | −.0753*** (−2.6858) | −.0076 (−.0221) | −.0977*** (−2.9977) |
| DEBT | .3273*** (16.9233) | .0263*** (4.7676) | −.3239*** (−3.6108) | .3572*** (17.4938) | .4847** (1.9964) | .4173*** (18.0948) |
| BSIZE | −.0406** (−2.3844) | .0013 (.2646) | .0697 (.8968) | −.0489*** (−2.7706) | .1885 (.8836) | −.06*** (−2.9621) |
| INTEN | .0002 (.5068) | .0012* (1.7222) | .0032 (1.4605) | .0003 (.5267) | .0008 (.1403) | .0001 (.0978) |
| TAT | −.0108 (−1.042) | .0137*** (3.393) | −.0619 (−1.2918) | −.0229** (−2.1051) | −.0377 (−.2884) | −.0231* (−1.8593) |
| SHARE | −.0006 (−1.1707) | −.0002 (−1.4753) | .0076*** (3.4701) | −.0003 (−.6354) | −.0019 (−.3188) | −.0008 (−1.4321) |
| DUAL | .0235*** (2.6229) | .0010 (.4241) | .1110*** (2.7042) | .0264*** (2.8227) | .1636 (1.4744) | .0220** (2.0894) |
| constant term | −.0343 (−.1938) | .1119** (2.5666) | 23.8259*** (28.8214) | .7900*** (3.5997) | 17.688*** (7.2354) | .3953* (1.6816) |
| Observations | 2821 | 4589 | 2667 | 2667 | 2355 | 2355 |
| R2 | .1410 | .1290 | .2848 | .1752 | .1338 | .1767 |

*** p < .01, ** p < .05, * p < .1.

Column (4) in Table 5 shows that the impact of the synergistic transformation of digitization and greenization on the R&D investment of non-high-tech SMEs is −0.0073. The impact of R&D investment on enterprise risk-taking is −0.0059, which is significant at 1% level. It indicates that synergistic transformation of digitization and greenization of non-high-tech SMEs can improve enterprise risk-taking by reducing R&D investment. The main reason is that digitalization in non-high-tech SMEs helps improve enterprises' organizational structure, production process, and technical level. Other than that, it integrates digital knowledge and green knowledge, improves the efficiency of R&D investment, reduces the level of R&D investment, and thus enhancing enterprise risk-taking.

### Endogeneity test

In this paper, the Hausman test determines whether the model has endogeneity problems. This paper draws on the study of Luo [25]. It uses the degree of synergistic transformation of digitization and greenization in the city where the enterprise is located (CITYSDG) as an instrumental variable. According to the first law of geography, everything is related to everything else. However, close things are more related than far away [69]. Therefore, the degree of CITYSDG affects SDG but does not directly affect enterprise risk-taking. Moreover, the data at the regional level are complex. They can be

affected by the characteristics of enterprises and meet the exclusion constraints. Moreover, the underidentification test (P value = 0.0000) and weak identification test (F = 23.01 > 10% critical value) are conducted in this paper, and the results show that CITYSDG is a qualified instrumental variable. Hausman test results show that Prob>chi2 = 1.0000, indicating that the fixed effect model used in this paper does not have significant endogeneity problems.

### Robustness test

**Changing control variables.** In this paper, control variables were changed to test the robustness of the main study conclusions (as shown in Table 6). Section A of Table 6 shows the results without adding control variables. Section B shows the results of controlling for only financial indicators (including ROE, DEBT, INTEN, and TAT). Section C lists the results of controlling for only the corporate governance indicators (including AGE, BSIZE, SHARE, and DUAL). Column (1) lists the impact of SDG on RISK, and the coefficients are all significantly positive, which shows that the synergistic transformation of digitization and greenization significantly promotes enterprise risk-taking. Columns (2) and (3) present the mediating effect of CASH. Columns (4) and (5) show the mediating effect of RD. The results validate SMEs' synergistic transformation of digitization and greenization, which promotes enterprise risk-taking by reducing cash holdings and R&D investments. This paper's main conclusions have not changed according to the above results.

**Change the measurement of main variables.** To test the reliability of the conclusions further, this paper changed the measurement of the main variables (as shown in Table 7). The results of changing the calculation of dependent variables are demonstrated in section A of Table 7. Referring to the practice of Shahzad [70], Yung & Chen [71], the standard deviation of the three-year adjusted ROE is used to measure enterprise risk-taking (RISK*) without changing the independent variables and control variables. The results of changing the calculation of independent variables are shown in section B of Table 7. This paper uses the data of all listed companies in China from 2012 to 2022 to calculate the SMEs' synergistic transformation of digitization and greenization (SDG*) without changing the dependent and control variables. Column (1) shows the impact of SMEs' synergistic transformation of digitization and greenization on enterprise risk-taking, and the coefficients are all significantly positive. Columns (2) and (3) present the mediating effect of CASH. Columns (4) and (5) show the mediating role of RD. These results validate that the main conclusions have not changed.

## Discussion

Based on the synergetics, information asymmetry, and agency theories, this paper explores the impact of SMEs' synergistic transformation of digitization and greenization on enterprise risk-taking. It has been found that SMEs' synergistic transformation of digitization and greenization can significantly promote enterprise risk-taking. Referring to the research of Wen [54], the mediating effect of cash holdings and R&D investment is supported. These conclusions reveal the economic consequences and mechanism of SMEs' synergistic transformation of digitization and greenization, which are significantly different from previous studies.

This paper holds that SMEs' synergistic transformation of digitization and greenization significantly impacts enterprise risk-taking, and cash holdings and R&D investment have a significant negative mediating effect. In previous studies, some scholars also emphasized the economic consequences of synergistic digitization and greenization. For example, Lv and Chen [38] analyzed the synergistic effect of digitization and greenization in different regions to help promote the efficiency of green economy. In this paper, the results are verified and extended from the microscopic perspective.

Unlike the study of Lv and Chen [38], this paper measures the impact of SMEs' synergistic transformation of digitization and greenization on enterprise risk-taking from the enterprise level, which is more valuable for guiding digital transformation and green transformation together and improving investment efficiency. There is still a lack of research on the economic consequences of SMEs' synergistic transformation of digitization and greenization. The existing literature finds that digital transformation promotes enterprise risk-taking using Chinese samples [8, 15, 24, 25}. It has also been discovered that enterprise green behavior can promote risk-taking [10,26]. The conclusions of this paper are significantly different from those studies.

**Table 6. Change control variable.**

**Section A: No control variables**

| | (1) RISK | (2) CASH | (3) RISK | (4) RD | (5) RISK |
|---|---|---|---|---|---|
| SDG | .0050** (2.1891) | −.1070*** (−6.7957) | .0031 (1.3118) | −.0808** (−2.5455) | .0043* (1.7504) |
| CASH | | | −.0195*** (−10.1366) | | |
| RD | | | | | −.0027*** (−2.6528) |
| constant term | .0087 (.1261) | 23.3877*** (48.4358) | .4657*** (5.4699) | 17.9385*** (17.2254) | .0864 (1.0439) |
| Observations | 7410 | 6995 | 6995 | 6741 | 6741 |
| R2 | .0313 | .2783 | .0484 | .1755 | .0326 |

**Section B: Only financial indicators controlled**

| | (1) RISK | (2) CASH | (3) RISK | (4) RD | (5) RISK |
|---|---|---|---|---|---|
| SDG | .0070*** (3.1561) | −.1086*** (−6.9148) | .0054** (2.3505) | −.0776** (−2.4419) | .006** (2.5261) |
| CASH | | | −.0167*** (−9.0281) | | |
| RD | | | | | −.0035*** (−3.5949) |
| ROE | −.0044*** (−11.2519) | .0053** (1.9912) | −.0043*** (−11.0571) | .0027 (.5068) | −.0042*** (−10.6534) |
| DEBT | .1361*** (16.8857) | −.2163*** (−3.7284) | .1516*** (18.053) | .4155*** (3.3596) | .1787*** (19.4606) |
| INTEN | .0008** (2.4459) | .0001 (.0641) | .0008** (2.3086) | −.0066 (−1.4776) | .0006* (1.8883) |
| TAT | .0178*** (3.7595) | −.1866*** (−5.4491) | .0099** (1.9987) | .0423 (.6088) | .0129** (2.5055) |
| constant term | −.1425** (−2.1264) | 23.6483*** (48.7796) | .2461*** (2.9765) | 17.6384*** (16.8667) | −.0433 (−.5448) |
| Observations | 7410 | 6995 | 6995 | 6741 | 6741 |
| R2 | .0977 | .2845 | .1211 | .1774 | .1169 |

**Section C: Only governance indicators controlled**

| | (1) RISK | (2) CASH | (3) RISK | (4) RD | (5) RISK |
|---|---|---|---|---|---|
| SDG | .0048** (2.1234) | −.1041*** (−6.7162) | .0031 (1.3027) | −.0749** (−2.3576) | .0042* (1.6849) |
| CASH | | | −.0192*** (−9.8236) | | |
| RD | | | | | −.0026** (−2.5313) |
| AGE | −.0019 (−.1751) | −.2411*** (−3.0673) | .0030 (.2494) | −.1527 (−.9393) | .0074 (.5873) |
| BSIZE | −.0103 (−1.3558) | .0384 (.7471) | −.0104 (−1.3312) | .2881*** (2.7313) | −.0099 (−1.208) |
| SHARE | −.0005*** (−2.8091) | .0163*** (12.7633) | −.0003 (−1.3717) | .0069*** (2.6334) | −.0005*** (−2.7028) |
| DUAL | .0086** (2.2567) | .1095*** (4.2195) | .0111*** (2.8141) | .1068** (2.0182) | .0077* (1.8801) |

*(Continued)*

**Table 6.** (Continued)

**Section A: No control variables**

| | (1) RISK | (2) CASH | (3) RISK | (4) RD | (5) RISK |
|---|---|---|---|---|---|
| constant term | .0740 (.9372) | 23.0367*** (41.8856) | .4971*** (5.2280) | 17.1780*** (14.4420) | .1225 (1.3043) |
| Observations | 7410 | 6995 | 6995 | 6741 | 6741 |
| R2 | .0337 | .3033 | .0503 | .1785 | .0349 |

*** p < .01, ** p < .05, * p < .1.

Firstly, based on the synergetics theory, this paper uses the Haken model for the first time to measure the degree of SMEs' synergistic transformation of digitization and greenization at the enterprise level. It expands the application field of the Haken model. Secondly, this paper analyzes the economic consequences of the synergistic transformation of digitization and greenization of smaller and more growing SMEs. This provides a new perspective for the government and relevant departments to understand how SMEs adapt to the digital and green economy. Thirdly, the existing research has not focused on the mediating effect of cash holdings and R&D investment. Sun [72] established that after enterprises' digital transformation, the changes brought by digital technology could alleviate information asymmetry, reduce agency conflicts, and reduce transaction costs, thus reducing the cash holding of enterprises. This paper argues that SMEs' synergistic transformation of digitization and greenization can reduce cash holdings through agency motivation and prevention motivation and thus increasing enterprise risk-taking, further extending Sun et al.'s findings [69]. Enterprise digital transformation contributes to innovation [15]. This paper extends the above findings. This paper states that SMEs' synergistic transformation of digitization and greenization can help realize the collaboration between the digital and green subsystems, improving innovation quality and promoting enterprise risk-taking while reducing R&D investment. Fourthly, in the heterogeneity analysis, this paper analyzes the impact of SMEs' synergistic transformation of digitization and greenization on enterprise risk-taking from the perspective of property rights and technological attributes.

These conclusions provide a new perspective and thinking for the government and relevant departments to formulate policies to promote digital and green transformation together. In conclusion, this paper extends the existing research. It has certain theoretical significance for scholars to measure the synergistic transformation of digitization and greenization and its economic consequences from the micro level. This paper also has a certain value for SMEs in understanding the impact of the synergistic transformation of digitization and greenization on long-term development. However, this paper also has some limitations.

Firstly, this paper does not discuss the lag effect of SMEs' synergistic transformation of digitization and greenization on enterprise risk-taking. Secondly, the moderating mechanism of the impact of SMEs' synergistic transformation of digitization and greenization on enterprise risk-taking is not involved in this paper, which may be the direction of further research in the future. Thirdly, this study takes enterprises listed on SME boards as samples, and its universality needs to be tested. In the future, we could further study the impact of the synergistic transformation of digitization and greenization on enterprise risk-taking using Western enterprises in other countries, especially SMEs in developing countries.

## Conclusions and suggestions

### Conclusions

This paper analyzes the impact of SMEs' synergistic transformation of digitization and greenization on risk-taking of enterprises listed on SME boards from 2012 to 2022. It draws the following conclusions: (1) SMEs' synergistic transformation of digitization and greenization significantly promotes enterprise risk-taking. In the development of the digital economy and the green economy, the synergistic transformation of digitization and greenization promotes orderly collaboration between

**Table 7. Change the measurement of main variables.**

**Section A: Change the calculation method of dependent variable**

| | (1) RISK* | (2) CASH | (3) RISK* | (4) RD | (5) RISK* |
|---|---|---|---|---|---|
| SDG | .0069*** (3.1476) | −.1049*** (−6.7817) | .0053** (2.3347) | −.0719** (−2.2637) | .0059** (2.5239) |
| CASH | | | −.0178*** (−9.538) | | |
| RD | | | | | −.0036*** (−3.6989) |
| ROE | −.0044*** (−11.4581) | .0055** (2.089) | −.0044*** (−11.3473) | .0027 (.505) | −.0043*** (−10.9403) |
| AGE | −.0288*** (−2.7884) | −.1928** (−2.4364) | −.0302*** (−2.6337) | −.2460 (−1.497) | −.0336*** (−2.788) |
| DEBT | .1421*** (17.5227) | −.1772*** (−3.0737) | .1589*** (18.9792) | .4494*** (3.5954) | .1877*** (20.4523) |
| BSIZE | −.0153** (−2.0896) | .0465 (.9080) | −.0151** (−2.0256) | .2743*** (2.6011) | −.0135* (−1.7409) |
| INTEN | .0009*** (2.7136) | .0011 (.4978) | .0008*** (2.6246) | −.0062 (−1.3763) | .0007** (2.127) |
| TAT | .0211*** (4.4623) | −.1635*** (−4.8488) | .0141*** (2.8734) | .054 (.7764) | .017*** (3.3365) |
| SHARE | −.0005*** (−2.6839) | .0161*** (12.6515) | −.0003 (−1.5355) | .0069*** (2.6533) | −.0005*** (−2.8012) |
| DUAL | .0076** (2.0743) | .107*** (4.1371) | .0097*** (2.5852) | .1085** (2.0511) | .0078** (1.9994) |
| constant term | −.0113 (−.1479) | 23.1001*** (42.074) | .3958*** (4.3727) | 17.1425*** (14.416) | .1063 (1.1971) |
| Observations | 7410 | 6995 | 6995 | 6741 | 6741 |
| R2 | .1059 | .3081 | .1335 | .1807 | .1287 |

**Section B: Change the calculation method of independent variable**

| | (1) RISK | (2) CASH | (3) RISK | (4) RD | (5) RISK |
|---|---|---|---|---|---|
| SDG* | .0065*** (3.0175) | −.1026*** (−6.7804) | .005** (2.2466) | −.0703** (−2.2654) | .0055** (2.3772) |
| CASH | | | −.0168*** (−8.8977) | | |
| RD | | | | | −.0035*** (−3.5538) |
| ROE | −.0044*** (−11.2637) | .0055** (2.089) | −.0043*** (−11.066) | .0027 (.5050) | −.0042*** (−10.6902) |
| AGE | −.0276*** (−2.6643) | −.1928** (−2.4363) | −.0283** (−2.4384) | −.2460 (−1.4971) | −.0326*** (−2.676) |
| DEBT | .1389*** (17.0943) | −.1772*** (−3.0735) | .1546*** (18.2566) | .4494*** (3.5954) | .1823*** (19.6414) |
| BSIZE | −.0152** (−2.0648) | .0465 (.9081) | −.0152** (−2.018) | .2743*** (2.601) | −.0138* (−1.7612) |
| INTEN | .0008** (2.3902) | .0011 (.4978) | .0007** (2.2981) | −.0062 (−1.3763) | .0006* (1.8335) |
| TAT | .0180*** (3.8028) | −.1635*** (−4.8488) | .0104** (2.1065) | .0540 (.7764) | .0133*** (2.5816) |
| SHARE | −.0005** (−2.5434) | .0161*** (12.6512) | −.0003 (−1.4847) | .0069*** (2.6532) | −.0005*** (−2.6827) |

*(Continued)*

**Table 7.** (Continued)

**Section A: Change the calculation method of dependent variable**

|  | (1) RISK* | (2) CASH | (3) RISK* | (4) RD | (5) RISK* |
|---|---|---|---|---|---|
| DUAL | .0088** | .1071*** | .0107*** | .1085** | .0087** |
|  | (2.383) | (4.1374) | (2.8036) | (2.0512) | (2.224) |
| constant term | .0213 | 22.7647*** | .3970*** | 16.9143*** | .1304 |
|  | (.2983) | (44.2115) | (4.5692) | (15.0751) | (1.5375) |
| Observations | 7410 | 6995 | 6995 | 6741 | 6741 |
| R2 | .1008 | .3081 | .1240 | .1807 | .1200 |

*** p<.01, ** p<.05, * p<.1.

the digital and green subsystems, changing enterprises' production modes. It helps enterprises establish unique competitive advantages, improve investment efficiency, and thus increasing enterprise risk-taking. (2) The mechanism analysis finds that SMEs' synergistic transformation of digitization and greenization helps reduce cash holdings and R&D investment and increases enterprise risk-taking. On the one hand, the synergistic transformation of digitization and greenization helps reduce SMEs' cash holdings through agency and precautionary motives. It encourages SMEs to collect superior resources for investment activities, thus improving enterprise risk-taking. On the other hand, the synergistic transformation of digitization and greenization helps SMEs concentrate on superior resources to carry out two-way collaboration between digital and green R&D investment. It reduces the amount of R&D investment, improves the efficiency of R&D investment, and enhances enterprise risk-taking. (3) Further research finds that, compared with state-owned SMEs, the synergistic transformation of digitization and greenization of non-state-owned SMEs has a more significant impact on enterprise risk-taking. Compared with high-tech SMEs, the synergistic transformation of digitization and greenization of non-high-tech SMEs has a more significant impact on enterprise risk-taking. These conclusions have practical significance for promoting the SMEs' synergistic transformation and improving enterprise risk-taking.

## Suggestions

Strong growth and innovation make SMEs important for China's economic development. Under the background of the development of the digital economy and green economy, in order to promote the SMEs' synergistic transformation of digitization and greenization and improve the level of enterprise risk-taking, this paper puts forward the following suggestions:

Firstly, local governments should formulate policies according to encourage SMEs synergistic transformation of digitization and greenization. Risk-taking is an important driving force for enterprises to achieve sustainable development. Local governments should formulate relevant policies in line with the characteristics of SMEs according to the development dynamics of their digital economy and green economy, such as financial subsidies, technical assistance, and other policies. It aims to help SMEs carry out the synergistic transformation of digitization and greenization, help enterprises transform and upgrade, and increase enterprise risk-taking.

Secondly, SMEs should improve the efficiency of capital utilization and reduce cash holdings. SMEs should use the synergistic system of digitization and greenization to establish the diversified information acquisition mechanism and build a digital platform to transmit green and sustainable development information to enterprises. It can establish a good image for enterprises, attract more external funds to deepen the synergistic transformation of digitization and greenization, alleviate the agency problem between organizations, reduce the level of cash holdings, and improve the efficiency of capital utilization to promote the level of enterprise risk-taking.

Thirdly, SMEs should improve their technological level and the efficiency of R&D investment. SMEs should play a full role in the synergy between the digital and green subsystems and promote the efficiency of R&D investment through

the deep integration of digital technology and green innovation. Using big data, Artificial Intelligence (AI), the Internet of Things, and other digital technologies to optimize the R&D process, SMEs could reduce trial and error costs. It improves the efficiency of R&D investment, accelerates the breakthrough of green technology, and finally achieves a synergistic development in production efficiency, environmental performance, and risk-taking, as well as builds core competitiveness in the wave of sustainable development.

## Author contributions

**Conceptualization:** Juan Wang.

**Data curation:** Jianxin Cui.

**Formal analysis:** Jianxin Cui.

**Funding acquisition:** Juan Wang.

**Methodology:** Juan Wang.

**Writing – original draft:** Juan Wang, Jianxin Cui.

**Writing – review & editing:** Juan Wang.

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
