## [Decision Letter · Decision Letter 0]

19 Mar 2025

PONE-D-24-44514Can SMEs synergistic transformation of digitization and greenization promote enterprise risk taking?PLOS ONE

Dear Dr. Wang,

Thank you for submitting your manuscript to PLOS ONE. After careful consideration, we feel that it has merit but does not fully meet PLOS ONE’s publication criteria as it currently stands. Therefore, we invite you to submit a revised version of the manuscript that addresses the points raised during the review process.

We look forward to receiving your revised manuscript.

Kind regards,

José Antonio Clemente Almendros, PhD

Academic Editor

PLOS ONE

Journal Requirements:

2. Please update your submission to use the PLOS LaTeX template. The template and more information on our requirements for LaTeX submissions can be found at http://journals.plos.org/plosone/s/latex .

3. In the online submission form, you indicated that your data is available only on request from a third party. Please note that your Data Availability Statement is currently missing [the name of the third party contact or institution / contact details for the third party, such as an email address or a link to where data requests can be made]. Please update your statement with the missing information.

4. Please ensure that you refer to Figure 1 in your text as, if accepted, production will need this reference to link the reader to the figure.

5. Thank you for stating the following in your manuscript:

“This paper is funded by “the Shandong Provincial Social Science Planning Research Project”, “A Study on the Impact of Corporate Digital Transformation on Risk Taking in Shandong Province (22CGLJ35)”

“This paper is funded by “the Shandong Provincial Social Science Planning Research Project”, “A Study on the Impact of Corporate Digital Transformation on Risk Taking in Shandong Province (22CGLJ35)”

Additional Editor Comments (if provided):

Dear Author(s)

In addition to the reviewers suggestions, I encourage you to include in your manuscript how you deal with potential endogeneity problems in your data and models.

Thank you

Reviewers' comments:

Reviewer's Responses to Questions

**Comments to the Author**

1. Is the manuscript technically sound, and do the data support the conclusions?

Reviewer #1: Partly

Reviewer #2: Yes

2. Has the statistical analysis been performed appropriately and rigorously? 

Reviewer #1: N/A

Reviewer #2: Yes

3. Have the authors made all data underlying the findings in their manuscript fully available?

Reviewer #1: Yes

Reviewer #2: Yes

4. Is the manuscript presented in an intelligible fashion and written in standard English?

Reviewer #1: No

Reviewer #2: Yes

5. Review Comments to the Author

Reviewer #1: This paper employs the Haken model to assess the synergistic transformation of digitization and greening in small and medium-sized enterprises (SMEs) and examines its impact on enterprise risk-taking. This paper is interesting. However, I have the following comments:

(1) The research model is simple and lacks in-depth discussion for their research. The abstract is overly focused on the background information, which fails to adequately highlight the novelty and significance of their research. The abstract should be improved to emphasize the study’s contributions and its broader implications.

(2) Literature Review and introduction lacks enough latest references to help the readers understand their research, the current version is difficult to raise concerns about the article’s originality and relevance.

(3) The knowledge gap of their research in Literature Review is not clear for reader, indicating that it is difficult to understand their contributions or to provide a forward-looking perspective. The authors should revise this part.

(4) The flowchart of the model is low quality which lacks a detailed description of the research process. The authors can add a paragraph to describe the main step of the method which may enhance readability of their model.

(5) Their policy implications do not provide specific recommendations based on the conclusions found in the paper's research, but rather broad policy implications. It means that the policy implications should have a close relationship with their results.

(6)The manuscript is recommended to be checked for English by a native speaker.

Reviewer #2: Can SMEs Synergistic Transformation of Digitization and Greenization Promote Enterprise Risk Taking?

The manuscript presents an important contribution to the discourse on SME transformation by integrating digitization and greenization. Given the increasing emphasis on sustainability and digital adaptation in business, the research is relevant to contemporary economic and management studies. The significance of the study is well-articulated, particularly in the context of China’s economic policies and SMEs’ role in sustainable development. The findings have practical implications for SME decision-making and policy development, reinforcing the study’s contribution to the field.

While the paper presents a valuable contribution, it requires minor revisions before publication:

1. Provide a clearer rationale for using the Haken model, including examples of its application in similar contexts.

2. While the study acknowledges prior research on digital and green transformations individually, it could benefit from a more extensive discussion of studies that have examined their synergistic effects. Discuss prior research on the combined impact of digitization and greenization more extensively.

3. The paper claims that reducing cash holdings and R&D investment enhances risk-taking, but this assumption may not hold universally. Additional robustness checks or alternative measures of risk-taking could strengthen the argument. Kindly offer alternative explanations for why cash holdings and R&D investment reductions lead to increased risk-taking.

4. Provide more transparency on data sources, keyword selection validation, and the paper does not indicate whether the datasets or analysis scripts will be made available for replication.

6. PLOS authors have the option to publish the peer review history of their article (what does this mean? ). If published, this will include your full peer review and any attached files.

**Do you want your identity to be public for this peer review?** For information about this choice, including consent withdrawal, please see our Privacy Policy .

Reviewer #1: No

Reviewer #2: No

---

## [Author Response · Author response to Decision Letter 1]

15 May 2025

Additional Editor Comments : In addition to the reviewers suggestions, I encourage you to include in your manuscript how you deal with potential endogeneity problems in your data and models.

The response of author

Thank you for the comments of the academic editor. Although the fixed effects model is used in this paper, there are still possible endogeneity problems. We have incorporated the academic editor's suggestions for additional endogeneity test. In order to determine the possible endogeneity problems, this paper uses the degree of synergistic transformation of digitization and greenization in the city where the enterprise is located (CITYSDG) as an instrumental variable, and uses the hausman test to verify whether the model has endogeneity. A section "Endogeneity test" �line 729-742�has been added to the "Empirical results and analysis" of the paper. It is expressed as follows:

Endogeneity test

In this paper, the Hausman test determines whether the model has endogeneity problems. This paper draws on the study of Luo [71]. It uses the degree of synergistic transformation of digitization and greenization in the city where the enterprise is located (CITYSDG) as an instrumental variable. According to the first law of geography, everything is related to everything else. However, close things are more related than far away [72]. Therefore, the degree of CITYSDG affects SDG but does not directly affect enterprise risk-taking. Moreover, the data at the regional level are complex. They can be affected by the characteristics of enterprises and meet the exclusion constraints. Moreover, the underidentification test (P value=0.0000) and weak identification test (F=23.01>10%10% critical value) are conducted in this paper, and the results show that CITYSDG is a qualified instrumental variable. Hausman test results show that Prob>chi2 = 1.0000, indicating that the fixed effect model used in this paper does not have significant endogeneity problems.

---

## [Decision Letter · Decision Letter 1]

22 Oct 2025

PONE-D-24-44514R1Can SMEs synergistic transformation of digitization and greenization promote enterprise risk taking?PLOS ONE

Dear Dr. Wang,

Thank you for submitting your manuscript to PLOS ONE. After careful consideration, we feel that it has merit but does not fully meet PLOS ONE’s publication criteria as it currently stands. Therefore, we invite you to submit a revised version of the manuscript that addresses the points raised during the review process.

**ACADEMIC EDITOR:** The paper needs a recheck of English grammar. Also, please improve paragraph transitions, and ensure consistent citation formatting throughout the manuscript

We look forward to receiving your revised manuscript.

Kind regards,

Valentina Diana Rusu, PhD

Academic Editor

PLOS ONE

Journal Requirements:

Reviewers' comments:

Reviewer's Responses to Questions

**Comments to the Author**

1. If the authors have adequately addressed your comments raised in a previous round of review and you feel that this manuscript is now acceptable for publication, you may indicate that here to bypass the “Comments to the Author” section, enter your conflict of interest statement in the “Confidential to Editor” section, and submit your "Accept" recommendation.

Reviewer #1: All comments have been addressed

Reviewer #3: All comments have been addressed

2. Is the manuscript technically sound, and do the data support the conclusions?

Reviewer #1: Yes

Reviewer #3: Yes

3. Has the statistical analysis been performed appropriately and rigorously? 

Reviewer #1: Yes

Reviewer #3: Yes

4. Have the authors made all data underlying the findings in their manuscript fully available?

Reviewer #1: Yes

Reviewer #3: Yes

5. Is the manuscript presented in an intelligible fashion and written in standard English?

Reviewer #1: Yes

Reviewer #3: Yes

6. Review Comments to the Author

Reviewer #1: I have reviewed the revised manuscript. The authors have carefully addressed all the concerns raised in the initial review.

Reviewer #3: The revised manuscript titled “Can SMEs’ Synergistic Transformation of Digitization and Greenization Promote Enterprise Risk Taking?” demonstrates substantial improvement and now reflects a technically sound and well-structured piece of empirical research. The authors have adequately incorporated the editor’s earlier suggestion regarding potential endogeneity by introducing a robust instrumental variable approach using the CITYSDG index and performing a Hausman and weak identification test. This significantly enhances the methodological credibility of the paper.

The manuscript is well organized, with a logical flow from theoretical background and hypothesis formulation to empirical modeling and discussion. The application of the Haken model to measure the synergy between digitalization and greenization is novel and theoretically grounded, offering a fresh analytical lens for understanding SME transformation dynamics. The data selection from 889 SMEs across 2012–2022, derived from the CSMAR database, is appropriate and clearly screened. The econometric framework—fixed-effects regression combined with mediation analysis—has been implemented rigorously, supported by robustness checks, heterogeneity tests, and adequate discussion of endogeneity concerns.

The statistical presentation is precise, and the results align with the hypotheses proposed. Tables and variable definitions are clearly explained, enhancing transparency and reproducibility. The discussion effectively connects empirical findings with theoretical expectations, particularly the mechanisms involving cash holdings and R&D investment.

Minor editorial improvements could enhance the paper’s readability—especially by refining the English grammar, improving paragraph transitions, and ensuring consistent citation formatting throughout the manuscript. These are stylistic rather than substantive concerns.

Overall, the paper provides meaningful empirical evidence and valuable policy implications for SME digital–green transformation. It meets the scientific and technical standards required for publication in PLOS ONE.

7. PLOS authors have the option to publish the peer review history of their article (what does this mean? ). If published, this will include your full peer review and any attached files.

**Do you want your identity to be public for this peer review?** For information about this choice, including consent withdrawal, please see our Privacy Policy .

Reviewer #1: No

Reviewer #3: **Yes: ** VIJAY AGRAWAL

---

## [Author Response · Author response to Decision Letter 2]

8 Nov 2025

Dear academic editor and reviewers:

Thank you for taking the precious time to review my manuscript. These valuable comments are very helpful for improving the quality of the paper. Over the past three weeks, I have revised the format of the paper and standardized the writing style words by words. These revisions are of great help to the quality of the paper. The responses to the academic editor are as follows:

To academic editor

Comments : The paper needs a recheck of English grammar. Also, please improve paragraph transitions, and ensure consistent citation formatting throughout the manuscript.

The response of author

Thank you for the comments of the academic editor. I revised the entire paper in terms of grammar, the connection between sentences and paragraphs, and the format of citation, to make all of these more in line with English usage. All revised sentence are marked in yellow in Revised Manuscript with Track Changes. Thanks a lot.

---

## [Editor Report · Decision Letter 2]

14 Nov 2025

Can SMEs synergistic transformation of digitization and greenization promote enterprise risk taking?

PONE-D-24-44514R2

Dear Dr. Wang,

We’re pleased to inform you that your manuscript has been judged scientifically suitable for publication and will be formally accepted for publication once it meets all outstanding technical requirements.

Kind regards,

Valentina Diana Rusu, PhD

Academic Editor

PLOS ONE
---

## [Editor Report · Acceptance letter]

PONE-D-24-44514R2

PLOS ONE

Dear Dr. Wang,

I'm pleased to inform you that your manuscript has been deemed suitable for publication in PLOS ONE. Congratulations! Your manuscript is now being handed over to our production team.

Kind regards,

on behalf of

Dr. Valentina Diana Rusu

Academic Editor

PLOS ONE